# Tactical Masters Athletes: BMI Index Classifications

**DOI:** 10.3390/sports10020022

**Published:** 2022-02-09

**Authors:** Marc Keefer, Joe Walsh, Kent Adams, Mike Climstein, Chad Harris, Mark DeBeliso

**Affiliations:** 1Athletic Department, Central Washington University, Ellensburg, WA 98926, USA; keefer5421@gmail.com; 2Sport Science Institute, Sydney 2000, Australia; jfrwalsh@gmail.com; 3Kinesiology Department, California State University, Monterey Bay, Marina, CA 93933, USA; kadams@csumb.edu; 4Faculty of Health, Southern Cross University, Lismore 4225, Australia; michael.climstein@scu.edu.au; 5Physical Activity, Lifestyle, Ageing and Wellbeing Faculty Research Group, University of Sydney, Sydney 2000, Australia; 6Metropolitan State University of Denver, Denver, CO 80204, USA; charr112@msudenver.edu; 7College of Health Sciences, Southern Utah University, Cedar City, UT 84720, USA

**Keywords:** USMC, Marines, body mass index

## Abstract

Understanding the nexus between aging, physical activity, and obesity has been a source of ongoing investigation. A considerable amount of research has focused on Masters athletes in this regard, suggesting a beneficial relationship between Masters sport participation and a healthy body mass index (BMI). Some consider Active Duty military personnel as tactical athletes. As such, it is of interest to determine if aging Active Duty military personnel (or Masters Tactical Athletes) might have a similar BMI as other Masters athletes (MA). As such, this investigation examined previously recorded data of Active Duty Enlisted United States Marines (*n* = 402, male, 46–50 years old). The BMI of the Marines was stratified into categories of: underweight, normal, overweight, and obese. The Marines obesity prevalence was compared to US adult males (40–59 years) as well as male North American MA who competed at the 2009 Sydney World Masters Games. The Marines obesity prevalence was significantly lower than US adult males (*p* < 0.001) and those MA that competed in softball (*p* < 0.001); however, it was similar to MA that competed in football, track/field, swimming, and volleyball (*p* > 0.05). The average Marine BMI = 26.7 kg/m^2^ was similar to MA who competed in football, swimming, and volley ball (*p* > 0.05); however, it was higher than MA who competed in track/field (*p* < 0.05) and lower than MA who played softball (*p* < 0.05). It should be noted that the average BMI for the Marines and all MA sport categories were classified as being overweight. Within the parameters of this investigation, Tactical MA (i.e., aging US Marines) enjoy a similar beneficial BMI as other North American MA.

## 1. Introduction

It has been established that obesity is linked to grave health risks [1,2,3,4]. Global obesity prevalence continues to grow [4]. Likewise, the prevalence of obese and severely obese individuals has demonstrated a nearly 20-year increasing trend among adults in the United States (US) [5]. The prevalence of obesity among US adults is most pronounced among males aged 40–59 as metriced by body mass index (BMI) [5].

A number of the co-authors of this manuscript have investigated the “nexus between aging, physical activity and obesity” among competitive Masters athletes [6,7,8,9,10,11,12,13,14]. One such investigation focused on North American Masters athletes (MA) who competed at the Sydney 2009 World Masters Games [10]. The male North American MA (age: 52.6 ± 9.1 years, football, track/field, swimming, volleyball, and softball) all demonstrated a significantly lower prevalence of obesity when compared to that reported in the general population [10]. The salience of the study being that a key index of health, namely obesity as a health risk factor, is far lower in incidence among the study cohort of North American MA.

Recently, the term tactical athlete has been used to refer to service professionals of any age, gender, and race who are firefighters, law enforcement, emergency responders, and military who are physically and mentally capable of withstanding all of the physical hardship associated with their respective duties (Figure 1) [15]. The United States Marine Corps (USMC) defines a tactical athlete as any Marine who trains for combat readiness using a comprehensive athletic approach [16]. The tactical athlete incorporates strength, power, speed, and agility to help ensure their fitness level meets the requirements for combat readiness [16]. With that said, it may be reasonable to consider Active Duty Marines 40–50 years of age as tactical MA.

Given the aforementioned health benefits exhibited by North American MA regarding obesity, the authors of the current manuscript sought to determine if aging US Marines (tactical MA) exhibited a similar favorable prevalence of obesity as other North American MA. Specifically:the obesity incidence of US male adults 40–59 years was compared to that of Active Duty Enlisted US Marines 46–50 years of age;the obesity incidence of North American MA was compared to that of Active Duty Enlisted US Marines 46–50 years of age; andthe average BMI was compared between Active Duty Enlisted US Marines 46–50 years of age and North American MA.

## 2. Materials and Methods

### 2.1. Participants

This study was an examination of pre-existing data as collected by the USMC and archived in the USMC Operational Data Store Enterprise System and is a continuation of the Keefer et al. and Keefer et al. [17,18] investigations. The participant sample examined in the current study was comprised of 402 male, Active Duty Enlisted United States Marines between 46–50 years of age, as this age range was consistent with MA. The data examined was from USMC fitness records collected during the period of between 1 January 2017 to 8 December 2018. “The study was conducted according to the guidelines of the Declaration of Helsinki, and approved by the Institutional Review Board of Southern Utah University (#06-022019a, approved 2 June 2019)”. Likewise, “This research was carried out fully in accordance to the ethical standards of the International Journal of Exercise Science” as described previously [19].

### 2.2. Analysis

The mean and standard deviation (SD) were calculated for the Marine’s demographic variables [age, height, and body mass). Body mass index was calculated as body mass (kgs)/(height (m)^2^). The BMI scores were then sorted into the respective BMI (kg/m^2^) classifications of: underweight < 18.5, normal = 18.5–24.9, overweight = 25.0–29.9, and obese ≥ 30.0. A Chi-square test was used to compare the Marine’s obesity prevalence with that of US male adults 40–59 years of age. The US incidence of obesity prevalence has been reported as 46.4% for male adults 40–59 years of age [5]. Likewise, the prevalence of obesity among the Marines was compared with the prevalence of obesity as reported for North American Masters athletes who competed at the 2009 Sydney World Masters Games 2009.

A one sample t-test was used to compare the average Marine’s BMI with the upper limit of the overweight BMI classification (BMI = 29.9). An *a priori* power analysis was conducted with G*POWER 3.1.9.2 (Universitat Kiel, Germany) software with regard to conducting a one-sample t-test. In order to achieve a medium effect size of ES = 0.50 with a statistical power 1-β = 0.80 [two- tailed, and α = 0.05), 35 participants were required. The sample size examined in the current study consisted of *n* = 402 Marines.

The mean BMI was compared between the Marines and the other sport categories with a one-way ANOVA and subsequent Bonferroni post hoc t-tests. Significance was considered as α ≤ 0.05.

Statistical analysis were carried out with MS Excel 2013, which included spreadsheets developed by McDonald [20]. The Excel spreadsheets were peer examined for accuracy as recommended by AlTarawneh at al. [21].

## 3. Results

Descriptive information of the Marines are presented in Table 1 (*n* = 402). Table 2 provides the percentage break down of the Marine’s BMI classifications. Also contained in Table 2 are prevalence of BMI classifications of North American MA who competed in the 2009 Sydney World Masters Games.

The mean BMI for US Marines averaged 26.7 kg/m^2^, considered as overweight and considered significantly lower than the cut-off of 29.9 (*p* < 0.001). The prevalence of obesity among US Marines was significantly lower than that observed in the US male adult population 40–60 years of age (*p* < 0.001). It is worth mentioning that no Marine BMI reached 36, noting that severe obesity is considered as a BMI ≥ 40.0 [5].

The prevalence of obesity among US Marines was significantly lower than MA that competed in softball (*p* < 0.001), but was similar to MA that competed in football, track/field, swimming, and volleyball (*p* > 0.05) (see Table 2 and Figure 2).

The average Marine BMI was similar to MA who competed in football, swimming, and volley ball (*p* > 0.05), but was higher than athletes who competed in track/field (*p* < 0.05, ES = 0.74), and lower than athletes who played soft ball (*p* < 0.05, ES = 0.57) (see Table 2 and Figure 3).

The kurtosis and skewness of the BMI scores for the Marines and MA were examined, and no evidence for lack of normality were detected for any of the groups.

## 4. Discussion

The purpose of this investigation was to examine the BMI classification of aging Active Duty Enlisted Marines (Masters Tactical Athletes) in comparison to other MA as well as the US general population of 40–59-year-old men. It was hypothesized that the prevalence of obesity among aging Marines would be similar to other MA and less than that found in the general population. Further, it was hypothesized that the average BMI of the aging Marines would be similar to other MA. The results of the data analysis were mixed in regard to the research hypotheses.

The prevalence of obesity among the aging Marines was similar to that of the MA with the exception of soft ball players and significantly lower than that found in the US general population of 40–59-year-old men. The average aging Marine BMI was similar to MA who competed in football, swimming, and volley ball, but was higher than athletes who competed in track/field, and lower than athletes who played soft ball. With that said, the average BMI of the aging Marines and all of the MA were classified as Overweight (BMI = 25.0–29.9).

The average BMI of the aging Marines of the current investigation (BMI = 26.7 ± 2.3) was similar to that reported in other investigations of military personnel [17,18,22,23]. The reported BMI of military personnel in prior investigations includes: Active-duty Enlisted male Marines (*n* = 19678, 22.5 ± 1.3 years, BMI = 25.4 ± 2.7) [17], Active-duty female Enlisted Marines (*n* = 554, 22.7 ± 1.4 years, BMI = 23.6 ± 2.1) [18], Army male ROTC cadets (*n* = 145, 21.6 ± 2.9 years, BMI = 23.9 ± 2.9) [22], and Officer and Enlisted male United States Army National Guard Soldiers (*n* = 655, age not reported, BMI = 27.1 ± 3.9) [23]. The average BMI reported in the aforementioned studies would suggest either a normal or overweight classifications for military personnel. Given the rigorous physical training of military personnel, it may be reasonable to suspect that additional muscle mass is contributing to the overweight classification of many military personnel.

The average BMI of both the aging and younger male Marines ranks towards the low end of the overweight classification (BMI = 25.0–29.9). While it is not known if the BMI of the aging Marines is remaining stable due to BMI drift [24], we suspect that the associated physical training of the aging Marines is maintaining muscle mass that would be otherwise lost due to aging and/or sedentary behavior. However, there may be an additional factor to consider.

In 2008, the USMC established the Marine Corps Body Composition and Military Appearance Program (MCBCMAP) [25]. The objective of the MCBCMAP was to establish healthy weight and body composition standards, and to ensure all Marines present a suitable military appearance. The MCBCMAP establishes, evaluates, and enforces compliance with optimal weight, body composition, and military appearance standards. Marines who do not present a suitable military appearance are required to take all necessary action to improve their appearance within the prescribed timelines or can be formally assigned to the MAP for weight redistribution, vice loss, or formally assigned and provided appropriate resources, counseling, and unit diary entrees (MCO 6110.3A w/CH3) [25]. It is possible that an outcome of the MCBCMAP is the relatively stable BMI exhibited by the aging Marines.

While the lower prevalence of obesity among the aging Marines in this investigation is promising, it is not necessarily indicative of the impact that obesity is having on military forces in general. Currently, there is a “crisis” regarding the rising number of US citizens considered “Unfit to Serve” due to obesity, which is harming the armed-force’s ability to recruit and replenish necessary military personnel [26].

The Centers for Disease Control and Prevention (CDC) reports that between 2011 and 2015, obesity rose by 73% in Active Duty service members [26]. Furthermore, the CDC reported that obese Active Duty soldiers were one-third more likely to suffer a musculoskeletal injury while on deployment. Collectively, the financial obesity-related health care burden for the Department of Defense is estimated at $1.5 billion per annum [26].

Reyes-Guzman and colleagues conducted a 13-year prospective study regarding obesity among Active Duty US military personnel (male, female, Enlisted, warrant Officers, Officers, Army, Navy, Marines, and Air Force) [27]. They reported the collective prevalence of overweight and obesity increased from approximately 50 to 61 percent over the study duration, which was primarily due to an increase in obesity. Interestingly, the Marines had the lowest prevalence of obesity of any of the military branches at any point in the study, and at the study’s conclusion was reported as 6.2% (similar to that reported in the current investigation). Shiozawa et al. [28] investigated the prevalence of overweight and obesity in 467,000 US Active Duty male Enlisted and Officer personnel, with 70 percent being overweight or obese. Specifically, more than one-half (51.2%) of participants were overweight and 19.7 percent were obese. Further, the soldiers with obesity had a disproportionately greater number of medical complaints, including musculoskeletal, mental health, ear nose and throat, and endocrine. Based on the aforementioned investigations, it is clear that there is an increase in the prevalence of obesity among military personnel and that it is having a clear impact on operational readiness. However, based on the findings of Reyes-Guzman et al. [27] and the current study, it appears that the Marines are being impacted the least by an obesity crisis.

The primary limitations of the current study relate to the comparisons of populations being made. The sample of aging Marines in the current study was *n* = 402, which allowed for an acceptably powered statistical analysis. However, the age range of the Marines was 46–50 years, while the comparison US male general population’s age range was 40–59 years. Further, the male North American MA comparison group’s average age was 52.6 ± 9.1 years. The differences in the aforementioned comparison group ages should be kept in mind while interpreting the results of the current investigation. Likewise, it should be noted that while BMI is an index of health, it is not an assessment of body composition, such as the dual energy X-ray absorptiometry, noting that the deuterium dilution method is the preferred [29].

The current study focused on Active Duty Enlisted aging male Marines. Future studies should examine obesity prevalence among male and female Marine Officers as well as military personnel in the other branches of the armed forces at a more granulated level then did Reyes-Guzman et al. [27], Shiozawa et al. [28], and the current investigation. Likewise, longitudinal studies following Enlisted cohorts over time with respect to obesity prevalence may help pinpoint when preventative interventions could be best implemented.

This investigation examined the BMI classification of aging male Active Duty Enlisted US Marines (masters tactical athletes) in contrast to other MA as well as the US general population of 40–59-year-old males. The prevalence of Marines classified as obese was lower than that found in the general population and similar to other MA. The findings were then discussed in the broader sense with regard to the growing impact of obesity prevalence among US military personnel. The salience of the study being that a key index of health, namely obesity as a health risk factor, is far lower in incidence among the aging Marines (or tactical MA) and similar to that of North American MA. We believe the results of the current study may further the understanding of the nexus between aging, physical activity, and obesity and serve to inform military leadership.

## Figures and Tables

**Figure 1 sports-10-00022-f001:**
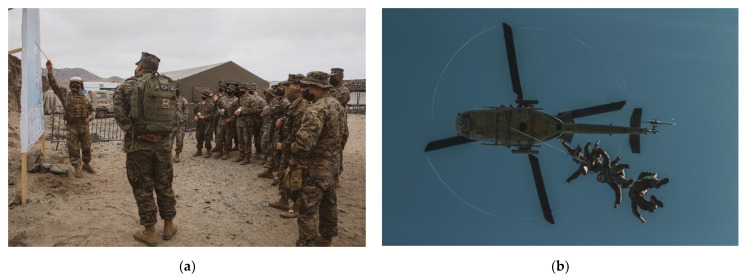
(**a**) Extensive mental and physical preparation provides the core foundation of every US Marine; (**b**) Tactical operations are a central function of a US Marine. Public Domain: www.dvidshub.net, accessed on 7 January 2022).

**Figure 2 sports-10-00022-f002:**
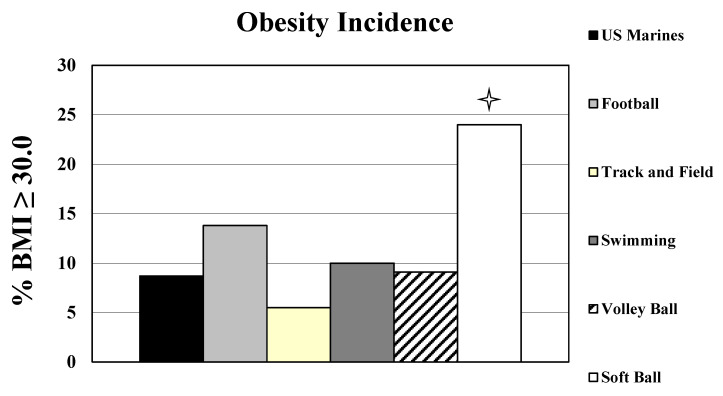
Obesity incidence for the Masters Athletes and US Marines. Star means softball players had a significantly higher obesity incidence than US Marines *p* < 0.05. Note: Obesity incidence of US population of males aged 40–59 years is 46.4%.

**Figure 3 sports-10-00022-f003:**
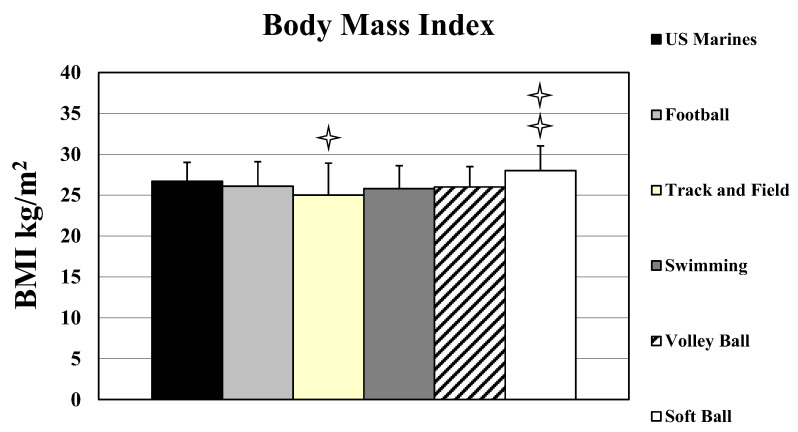
Average BMI for the Masters Athletes and US Marines. Double stars mean softball players had a significantly higher BMI than the US Marines *p* < 0.05. Star means Track and Field Athletes had a significantly lower BMI than the US Marines.

**Table 1 sports-10-00022-t001:** USMC Descriptive Information.

Marines	Age (y)	Height (m)	Mass (kg)	BMI (kg/m^2^)
Male (*n* = 402)	47.2 ± 1.2	1.76 ± 0.07	83.2 ± 10.0	26.7 ± 2.3

**Table 2 sports-10-00022-t002:** Body mass index and classification for multiple Master’s sports events.

	BMI (kg/m^2^)				
Context	Underweight % BMI < 18.5	Normal % BMI = 18.5–24.9	Overweight % BMI = 25.0–29.9	Obese BMI ≥ 30.0	Average BMI
US Marines ^1^ *n* = 402	0.0	18.2	73.1	8.7 *	26.7 ± 2.3
Football ^2^ *n* = 29	0.0	44.8	41.4	13.8 *	26.1 ± 3.0
Track & Field ^2^ *n* = 73	1.4	0.0	93.2	5.5 *	25.0 ± 3.9 ***
Swimming ^2^ *n* = 20	0.0	40.0	50.0	10.0 *	25.8 ± 2.8
Volley Ball ^2^ *n* = 33	0.0	36.4	54.5	9.1 *	26.0 ± 2.5
Soft Ball ^2^ *n* = 25	0.0	12.0	64.0	24.0 *^,^**	28.0 ± 3.0 ***

^1^ USMC Active Duty Enlisted personnel 46–50 years of age considered as Tactical Athletes. ^2^ Data courtesy of DeBeliso et al. 2014 North American Masters athletes BMI prevalence World Games 2009. * *p*< 0.05, significantly lower than U.S. obesity prevalence. ** *p* < 0.05, significantly higher than US Marines obesity prevalence. *** *p* < 0.05, significantly different from US Marines mean BMI.

## Data Availability

Data available on request due to restrictions. The data presented in this study is the property of the USMC. Interested parties may contact the corresponding author who will provide the appropriate USMC contact personal.

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
