# Peer review of "Tactical Masters Athletes: BMI Index Classifications"

_sports, 2022, doi:10.3390/sports10020022_

Round 1
Reviewer 1 Report
A generally well written manuscript describing a straightforward analysis of pre-existing data. I have some points that the authors should consider.
Overall, the authors link BMI with obesity, a condition of excess adiposity, i.e., a change in body composition. However, it is well recognized that BMI can be a poor reflection of body composition or metabolic function. This does not preclude BMI being used as an index but the point should be made that ideally measurement of BC using reference methods, e.g. deuterium dilution or secondary methods such as BIA should be used.
Specific points.
Page 2, section 2.1 I am not familiar with the US military but does “active duty enlisted” actually mean serving personnel not those on reserve? If so, as I believe is the case, this presumably includes personnel in all aspects of the Marine Corp, including those primarily on desk duties. Do the authors have any information on their actual physical activity status? If so this would be useful to include. Also it is not clear to me why only those aged 40-50 are included. I assume that the Marine Corp has many (a majority?) of younger personnel. Please justify more clearly the interest in this older age cohort.
Section 2.2 and Stats in general. Please verify that the data were all normally distributed and hence mean and SD are acceptable. If not, then present as median and IQR with appropriate non-parametric testing.
I am also a little surprised that an old version of Excel is used for statistical analysis. Excel is widely acknowledged as less than ideal for scientific analysis. Why was an appropriate package such as SPSS not used? If it is a matter of cost, ten some excellent freeware packages would do the job such as Jamovi or JASP. If Excel is to be used then I would highly recommend the use of the free statistical add in, “Real Statistics”. I am sure that your analysis is correct but for the future “professional” statistical analysis would be preferable.
What type of ANOVA was used? The data could be considered as a two factor (BMI stratum and Sport) analysis. This would allow consideration of any interaction.
Figure 1, while interesting is not particularly germane to the core of the paper and I suggest deletion.
Table 1. Y rather than yrs and kg not kgs.
Table 2 Provide units for BMI.
Discussion is acceptable. The authors mention as a limitation the lack of an age-comparable general population. I would suggest that NHANES data could supply this. It is publicly available for download.

Reviewer 2 Report
The manuscript deals with a topic of great interest to the scientific and military community, however I have some important perplexities. Considering the long period taken into consideration (1/1/2017 - 8/12/2018) I consider the results and analyzes reported to be decisively poor. It seems that the authors want to divide the study into many small parts. Despite this, since I consider the topic interesting, I invite the authors to restructure the manuscript by providing more detailed analyzes and results (for example, it could be possible to make correlations between BMI and hours of training and compare it between different sports activities) and perhaps to do even graphs that help readers understand the results more quickly. I hope I will be able to read a new version of your manuscript as soon as possible, and I hope it does not know a short communication.
Author Response
Please see the attachement.

Round 2
Reviewer 1 Report
The authors have satisfactorily addressed the issues raised in my original review.
Reviewer 2 Report
The manuscritp is suitale for publication